# Contrasting Responses and Phytoremediation Potential of Two Poplar Species to Combined Strontium and Diesel Oil Stress

**DOI:** 10.3390/plants12112145

**Published:** 2023-05-29

**Authors:** Ziyan Liang, Hanyong Zeng, Jijun Kong

**Affiliations:** 1Experimental Testing Team of Jiangxi Geological Bureau, Nanchang 330002, China; liangziyan725@163.com; 2Chengdu Institute of Biology, Chinese Academy of Sciences, Chengdu 610041, China; zhy2023520@163.com; 3Yunnan Laboratory for Conservation of Rare, Endangered & Endemic Forest Plants, National Forestry and Grassland Administration, Yunnan Academy of Forestry and Grassland, Kunming 650201, China

**Keywords:** combined stress, cross tolerance, diesel oil, strontium, physiological response, phytoremediation, *Populus*

## Abstract

The soil pollution caused by diesel oil and heavy metals has become an increasingly serious environmental issue, with negative global-scale impacts. The remediation of contaminated soil requires special attention, in which phytoremediation has emerged as an ecofriendly solution. However, the response of plants to the combined stress of diesel oil and heavy metals remains largely unknown. In this study, the aim was to investigate the potential of *Populus alba* and *P. russkii* for phytoremediation by examining their response to combined diesel oil and heavy metal stress. In a greenhouse experiment using soil contaminated with 15 mg kg^−1^ of diesel oil and varying concentrations of Sr (0, 10, or 100 mg kg^−1^), we studied the physiological and biochemical changes, as well as the Sr absorption, of *P. alba* and *P. russkii*. The results showed that at high concentrations of Sr and diesel oil, the growth of both species was substantially inhibited, but *P. alba* exhibited higher resistance due to its higher antioxidant enzyme activities and increased accumulation of soluble sugar and proline. Additionally, *P. alba* concentrated Sr in the stem, whereas *P. russkii* accumulated Sr in the leaf, exacerbating its negative effects. Diesel oil treatments were beneficial for Sr extraction due to cross-tolerance. Our findings indicate that *P. alba* is more suitable for the phytoremediation of Sr contamination due to its superior tolerance to combined stress, and we identified potential biomarkers for monitoring pollution. Therefore, this study provides a theoretical basis and implementation strategy for the remediation of soil contaminated by both heavy metals and diesel oil.

## 1. Introduction

The soil pollution caused by oil spills poses a serious problem due to the increasing global demand for diesel fuel [1], resulting in hydrocarbon toxicity in various ecosystems, including farmland, especially in developing countries [2]. Soil quality is crucial for food safety, and the adverse effects of heavy metals and other pollutants on crop quality threaten human health [3]. Heavy metal pollution is often associated with elements such as Ni, Cu, As, Cd, Sn, Zn, Pb, Hg, Mo, etc., among which Mo, Cd, Pb, Sr, and Mn are considered high-risk elements [4,5]. The soluble Sr in soil can be easily absorbed and accumulated by plants due to its chemical similarity to Ca, which can then enter the human body through the food chain, leading to negative impacts on the ecological environment [6].

Physical and chemical treatments are strategies that can be effectively applied for removing high levels of pollution, particularly in small areas. However, phytoremediation is considered one of the most suitable technologies for remediating contaminated water and soil, as it is environmentally friendly, potentially cost-effective, and does not require special equipment [7]. Additionally, phytoremediation and other biological methods can alter the chemical dynamics of metals in soil, leading to the efficient removal of metals from contaminated soil [8]. Fast-growing and heavy-metal-tolerant woody plants such as poplar are considered candidates for heavy metal phytoremediation [9].

Poplar is widely used in the management of polluted sites due to its ease of reproduction, rapid growth, deep roots, and high aboveground biomass [10]. Poplar has been widely employed in soil remediation. For instance, phytoremediation with *Populus tomentosa* has been used to remove petroleum hydrocarbons (PHCs) from the soil in northern climatic conditions [11], and poplar wool fiber possesses excellent absorption properties due to its unique chemical, physical, and microstructural properties, making it highly effective in absorbing heavy diesel oil [12]. Furthermore, evidence exists of sex-specific abilities of poplar in repairing contaminated soil. The combined exposure to Cd and Zn resulted in different responses between male and female *P. cathayana*, with male plants exhibiting stronger tolerance to and a higher accumulation ability of Cd at appropriate Zn levels [13]. Moreover, the asymmetry in sexual competition may play a role in regulating the population structure, spatial separation, and remediation potential of *P. cathayana* [14]. Under normal N supply, female plants showed higher leaf Cd accumulation and root–stem Cd transport compared with male plants, but under nitrogen deficiency, male plants exhibited superior Cd extraction ability [15]. Different species of poplars exhibit contrasting effects on soil remediation. European poplar and hybrid poplar clones are effective in remediating soils contaminated with polycyclic aromatic hydrocarbons (PAHs) and total petroleum hydrocarbons (TPHs). *Populus alba*, in particular, shows the strongest extraction and stability effects for several heavy metals, including Mg, Ca, Fe, Cu, and Na [16]. *P. russkii* shows strong tolerance to boron [17], but its tolerance to other heavy metals remains unclear. However, studies on the phytoremediation ability of poplars in the context of strontium (Sr)- and diesel-contaminated soils have been limited.

Plants, being sessile organisms, often simultaneously experience multiple stresses. The perception of abiotic stresses and the signaling mechanisms that trigger adaptive responses are crucial in determining the survival and reproduction of plants living in adverse environments [18]. The concept of cross-tolerance involves promoting tolerance to severe stress by simultaneously or sequentially subjecting plants to a milder stress [19,20]. Cross-tolerance can involve complex patterns of activation of stress-responsive pathways, leading to changes in morphology, anatomy, physiology, and biochemistry [21]. As such, cross-tolerance presents an important mechanism for plants to cope with multiple stress factors, including heavy metal and oil contamination. These findings highlight the potential of plant cross-tolerance as a mechanism for plants to cope with multiple stress factors, including heavy metal and oil contamination, and shed light on the intricate interplay between different stress responses in plants.

Therefore, in this study, *P. alba* and *P. russkii* were used as model woody plants to compare their physiological responses to combined stress from Sr and diesel oil, and to explore their potential for phytoremediation. Specifically, we compared their biomass accumulation, metal extraction, and physiological responses, including antioxidant enzyme activity and soluble substance contents. We aimed to (1) identify potential phytoremediation candidates with high metal extraction ability, particularly in the leaves and stems; (2) decipher the physiological basis of their tolerance to combined stresses, providing clues for future genetic manipulation to improve phytoremediation potential; and (3) investigate the existence of cross-tolerance between diesel and Sr contaminations, and to explore its potential application in phytoremediation.

## 2. Results

### 2.1. Growth of P. alba and P. russkii under Different Treatments

The growth of both *P. alba* and *P. russkii* was considerably inhibited by the high concentration (100 mg kg^−1^) of Sr, as evidenced by reduced plant height, leaf area, and biomass accumulation (Figure 1). The addition of diesel oil also negatively affected the biomass of both species, with *P. russkii* showing a more pronounced reduction compared with *P. alba*. Notably, the low concentration (10 mg kg^−1^) of Sr enhanced the growth of both species (Figure 1), leading to significant species × oil × Sr interactions, particularly for leaf area and plant height (Table 1).

### 2.2. Sr Concentration and Translocation among Different Tissues of P. alba and P. russkii

The Sr concentrations in the plant tissues increased with an increasing concentration of exogenously applied Sr in both species (Figure 2a–c). *P. russkii* tended to accumulate Sr in the leaves, whereas *P. alba* stored more Sr in the stems (Figure 2d–f). Considering tissue weight and Sr concentration, *P. alba* extracted more Sr from the soil than *P. russkii* (Figure 2g–i).

### 2.3. Activities of Antioxidant Enzymes among Different Treatments of P. alba and P. russkii

The Sr treatments induced oxidative stress, as indicated by significant increases in the malondialdehyde (MDA) contents in both species (*p* < 0.001). The MDA content was higher in *P. russkii* than in *P. alba*, suggesting more severe damage in *P. russkii* (Figure 3). Antioxidant enzymes, including ascorbate peroxidase (APX), peroxidase (POD), and superoxide dismutase (SOD), were stimulated by low-concentration Sr stress but inhibited by high-concentration Sr stress (Figure 3). Moreover, APX and SOD activities were significantly higher in *P. alba* than in *P. russkii*, whereas POD activity was lower in *P. alba* (Figure 3). Leaf enrichment in Sr was significantly positively correlated with APX activity in *P. alba* but negatively in *P. russkii*, whereas SOD and POD activities were negatively correlated with leaf Sr enrichment in both species (Figure 4).

### 2.4. Soluble Substance Content among Treatments of P. alba and P. russkii

Soluble substances, including sugar and proline, showed similar responses to strontium and diesel oil treatments (Figure 5). Their concentrations were increased by low-concentration Sr treatment but inhibited by high-concentration Sr treatment, with *P. russkii* consistently showing higher sugar and proline levels than *P. alba*.

### 2.5. Correlations among Growth and Physiological Parameters in P. alba and P. russkii

The heat map results revealed that most of the studied physiological parameters were strongly correlated with biomass accumulation in both species (Figure 6). In the PCA plot, different treatments aggregated together for *P. alba*, indicating higher resistance, whereas treatments were separated in *P. russkii* due to its lower tolerance (Figure 6). PC1 and PC2 explained 44.0% and 27.9% of the variation in *P. alba*, respectively (Figure 6). The biomass of *P. alba* was positively associated with SOD and APX activities, whereas the biomass of *P. russkii* was strongly related to MDA content, an oxidative damage parameter, and negatively related to SOD activity (Figure 6).

## 3. Discussion

Soil pollution caused by the combination of Sr and diesel oil presents a considerable challenge for both plants and humans. Thus, various studies have been conducted to identify suitable phytoremediation candidates and decipher the physiological bases for their superior adaptations.

### 3.1. Contrasting Responses of Two Poplar Species to Combined Stresses

High concentrations of Sr and diesel oil significantly inhibited the growth of *P. alba* and *P. russkii*. However, *P. alba* exhibited higher resistance due to its higher antioxidant enzyme activities and higher soluble sugar and proline accumulation. One possible mechanism was identified to be involved in the resistance to various types of stress: the activity of antioxidant enzymes such as superoxide dismutase (SOD), ascorbate peroxidase (APX), and catalase (CAT). The increased expression of these antioxidant enzymes under different types of stress may play a general role in enhancing the tolerance of plants to stress [22].

In a prior study, under high Cd concentrations, SOD activity was significantly lower than that of a control, whereas POD activity was higher than that of the control [23]. Rusin et al. [24] observed that plants growing on diesel-contaminated soil were characterized by lower SOD activity but higher POD activity and proline levels. Huang et al. [25] found that the oxidative stress in plants caused by diesel-treated soil was alleviated by adding selenium (0.5 or 1.5 mg/kg^−1^). In our study, under the lower stress caused by diesel oil and Sr, the activities of APX, SOD, and POD increased, and the concentrations of sugar and Pro increased, indicating that the combined stress of diesel and Sr resulted in a certain stimulation of the antioxidant enzyme system of the two poplar species; however, under high-concentration Sr stress, the antioxidant enzyme activities of the two species began to decrease (Figure 3 and Figure 4). In this study, the biomass of *P. alba* was positively associated with SOD and APX activities, whereas that of *P. russkii* was strongly related to MDA content, an oxidative damage parameter, and negatively related to SOD activity (Figure 6). This study contributes to our understanding of the physiological response mechanisms of these two poplar species under the combined stress of Sr and diesel oil and provides a theoretical basis and implementation strategy for remediating soil contaminated by heavy metals and diesel oil.

### 3.2. Higher Phytoremediation Potential for P. alba

*P. russkii* tended to accumulate Sr in its leaves, whereas *P. alba* stored more Sr in its stems (Figure 2d–f). Based on the tissue weight and Sr concentration, *P. alba* extracted more Sr from the soil than *P. russkii* (Figure 2g–i), making it a more suitable plant for the phytoremediation of Sr-contaminated soil. Gaudet et al. [26] considered two clones of black poplar (*P. nigra*) originating from different regions in Italy, and both clones exhibited a low Cd content in their leaves compared with other species. In contrast to the above results, under the combined stress of Sr and diesel oil, no significant difference in the concentrations of heavy metals in the roots, stems, or leaves of *P. alba* was found, and its enrichment ability substantially decreased with an increasing Sr concentration. However, the enrichment ability of *P. russkii* did not notably change with an increasing Sr concentration, possibly indicating that its ion exchange system was less damaged than that of *P. alba*. The effects of high Sr concentrations include cell membrane damage and impairment of the ion exchange absorption system, which can result in a decrease in the phytoremediation capacity of *P. alba*.

*P. russkii* exhibited an increased stem height and leaf number at low Sr concentrations, but this trend was reversed at high Sr concentrations, which led to a decrease in both parameters. Lu et al. [27] highlighted how excessive Cd can negatively impact the growth and development of roots, including root length (RL), root surface area (SA), specific root length (SRL), and root tip number. Similarly, the high concentration of Sr in this study inhibited the growth of both *P. alba* and *P. russkii*. This may be attributed to the detrimental interaction between toxic heavy metals and cells/molecules, leading to the generation of unnecessary reactive oxygen species (ROS), which can limit plant growth [28].

### 3.3. Cross-Tolerance between Strontium and Diesel Oil

Plant cross-tolerance refers to a phenomenon where exposure to one stress factor, such as heavy metal contamination, results in enhanced tolerance to other stress factors, including oil contamination [29,30]. This intriguing phenomenon has been documented in several studies, with gradual changes in environmental conditions inducing tolerance to extreme conditions in many plants. Studies are increasingly demonstrating the existence of cross-tolerance in plants, with tissue exposed to moderate stress inducing resistance to other stresses [31,32]. Stress factors inducing cross-tolerance include biotic factors as well as heavy metals, hypoxia, ultraviolet-B radiation, heat, high salt, drought, and cold stress [18,33]. In this study, diesel oil treatments were beneficial for Sr extraction due to cross-tolerance effects (Table 1, Figure 1 and Figure 2). Cross-tolerance may involve an intricate pattern of activation of stress-responsive pathways, leading to changes in morphology, anatomy, physiology, and biochemistry [21,34]. In the case of cold-treated *Arabidopsis* seedlings, lower levels of Pb were detected in the roots and shoots than in the control. This was associated, at least in part, with the activation of *AtPDR12* gene expression, which functions as a pump to exclude Pb(II)- and/or Pb(II)-containing toxic compounds from the cytoplasm to the exterior of the cell [35]. However, further research is required to determine the precise mechanisms underlying cold-mediated enhanced Pb(II) resistance, as well as other types of cross-tolerance [36].

Our study reported the existence of cross-tolerance to heavy metal and diesel oil stress, presenting a potential phytoremediation solution for combined toxicities. To strengthen these findings, experiments should be performed with plants both in a more advanced vegetative state and under longer periods of pretreatment exposure. Moreover, a key aspect, controlled elicitation, is necessary of the biochemical and molecular responses as well as the soil microbiome to find specific conditions under which plants show the desired responses, i.e., higher biomass accumulation and Sr extraction amount.

## 4. Materials and Methods

### 4.1. Plant Materials and Experimental Design

Samples from two contrasting species of *P. alba* and *P. russkii* were collected from Chengdu Botanical Garden (104°10′ E, 30°40′ N, 546 m.a.s.l.) of Sichuan Province, Southwest China in March 2021. After sprouting and growing for about a month, healthy cuttings of approximately equal height were selected and replanted into 5l plastic pots filled with homogenized soil. The selected properties of the soil used in this study were pH 7.0, 0.17% organic carbon, 0.03% organic nitrogen, 40 mg/kg of potassium, 1.8 mg/kg of phosphorus, 88.0% sand, 8.0% silt, and 4.0% clay. The plants were then grown under semi-controlled environmental conditions in a naturally lit greenhouse, with a temperature range of 18.0–32.0 °C and relative humidity range of 50–80%, and supplied with 800 mL of Hoagland’s solution every day. Hoagland’s nutrient solution consisted of 5 mM of Ca(NO_3_)_2_·4H_2_O, 5 mM of KNO_3_, 2 mM of MgSO_4_·4H_2_O, 1 mM of KH_2_PO_4_, 0.1 mM of EDTA-Fe, 46 µM of H_3_BO_3_, 9.1 µM of MnCl_2_·4H_2_O, 0.32 µM of CnSO_4_·5H_2_O, 0.76 µM of ZnSO_4_·7H_2_O, and 0.5 µM of H_2_MoO_4_·H_2_O.

After culturing for one month, the seedlings were exposed to Sr and oil stress. For oil treatment, 0 or 15 mg kg^−1^ of diesel was added. For Sr treatment, 0, 10, and 100 mg kg^−1^ of Sr was added in the form of SrCl_2_·6H_2_O. There were six treatments for oil, Sr combinations as 0/0 (CK), 0/10, 0/100, 15/0, 15/10, and 15/100. The experimental layout was completely randomized with three factors (two species, two oil concentrations, and three Sr concentrations). Sixty cuttings of each species were randomly allocated to stresses for three months. Each treatment included five replications and two cuttings per replication.

### 4.2. Determination of Enrichment Content of Sr

All seedlings were harvested at the end of the experiment and divided into leaves, stem, and roots. The samples were killed at 105 °C for 30 min, and then dried at 70 °C for 72 h to constant weight. After drying, the tissues were ground into powder. About 0.2 g of ground dry sample was added with 15 mL of mixed acid (perchloric acid:nitric acid = 1:3), heated in graphite furnace at 200 °C for 15 min to 20 min, until the solution became clear. The remaining digestion solution was fixed to 50 mL, shaken, and placed for 12 h. The content of Sr in each tissue was determined by atomic absorption spectrometry. The enrichment factor was calculated by the ratio of Sr concentration in leaf, stem, and root, respectively. The extraction amount was calculated as the total Sr extracted and stored in the plant tissues.

### 4.3. Determination of Antioxidant Enzymes and Lipid Peroxidation

The assay for SOD (EC 1.15.1.1) activity was based on inhibition of the photochemical reduction of NBT [4]. The reaction mixture contained 50 mmol/L of potassium phosphate, pH 7.8, 0.1 mmol/L of EDTA, 13 mmol/L of methionine, 75 µmol/L of NBT, 16.7 µmol/L of riboflavin, and enzyme source (approximately 25 µg of protein). Riboflavin was added last, and the reaction was performed under one 18W fluorescent lamp (350 µmol m^−2^ s^−1^) for 15 min. An illuminated blank without protein gave the maximum reduction of NBT, and the absorbance was measured at 560 nm. One unit of SOD is defined as the amount required to inhibit the photoreduction of NBT by 50%. The specific activity of SOD was expressed as units per mg^−1^ of protein. The activity of POD (EC 1.11.1.7) was determined by the guaiacol method, and the change in absorbance at 470 nm per minute was calculated as one unit of enzyme activity [15]. Ascorbate peroxidase (EC 1.11.1.7) was assayed as the decrease in absorbance at 290 nm (2.8 L mmol^−1^ cm^−1^) due to ascorbic acid oxidation [24]. The reaction mixture contained 50 mmol/L of potassium phosphate, pH 7.0, 1 mmol/L of sodium ascorbate, 2.5 mmol/L of H_2_O_2_, and enzyme source (approximately 50 µg of protein) in a final volume of 1 mL at 25 °C. Lipid peroxidation was determined as the concentration of TBARS, equated with malondialdehyde (MDA), where the products were quantified from the second derivative spectrum against standards prepared from 1,1,3,3-tetraethoxypropane [27]. The TBARS content (MDA) is expressed as nmol g^−1^ of dry weight (DW).

### 4.4. Quantification of Soluble Sugar and Free Proline Contents

The content of the soluble sugar seedlings was extracted with one gram of fresh sample homogenized in 5 mL of 3% sulphosalicylic acid on ice, and the homogenates were centrifuged at 6000× *g* for 20 min at 4 °C and determined using the soluble sugar detection kit (G0501F, Suzhou Grace Biotechnology Co., Ltd., Suzhou, China). For the proline contents, plant tissues were homogenized in 3% sulfosalicylic acid at 100 °C for 10 min. After centrifugation, the glacial acetic acid and ninhydrin were added to the supernatant to produce coloration. Then, free proline was extracted with toluene and measured using the spectrophotometer (UV-2102C, Unico Instrument, Shanghai, China) [4,24].

### 4.5. Data Analysis

For growth, Sr accumulation, and physiological responses, data were analyzed by a three-way analysis of variance (ANOVA), with species, oil, and Sr as the main fixed factors shown in Table 1. Individual differences among group means were determined by Tukey’s test of one-way ANOVA at a significance level of *p* < 0.05. All analyses were conducted with SPSS 18.0 (SPSS, Chicago, IL, USA). Before the analyses, data were assessed for normality and homogeneity of variances and log10 transformed when necessary to satisfy the assumptions of ANOVA. Bivariate relationships between foliar Sr concentration and antioxidant enzymes were examined with standardized major axis regressions, with the *sma* function of package ‘*smatr*’, in R version 3.2.3 (R Development Core Team 2015). Multivariate correlations among the physiological traits were further assessed by a principal component analysis (PCA). The average factor loading values of the different treatments were also compared to determine whether they were significantly separated along the PC axes.

## 5. Conclusions

This study demonstrated that the growth of *P. alba* and *P. russkii* was considerably inhibited under high concentrations of Sr and diesel oil in the soil. However, *P. alba* exhibited higher resistance than *P. russkii* due to its higher antioxidant enzyme activities and higher soluble sugar and proline accumulation. The Sr concentration was found to be higher in the stems of *P. alba* and in the leaves of *P. russkii*, which further intensified the negative effects of the heavy metal. The diesel oil treatments were observed to enhance the extraction of Sr from the soil due to cross-tolerance. Overall, our findings suggest that the two poplar species exhibit contrasting responses to combined Sr and diesel oil stress, with *P. alba* being more suitable for the phytoremediation of Sr contamination. In addition to highlighting potential biomarkers for pollution monitoring, this study also underscores the need for further investigations into the biochemical and molecular responses, as well as the soil microbiome, to enhance the phytoextraction ability of these plants.

## Figures and Tables

**Figure 1 plants-12-02145-f001:**
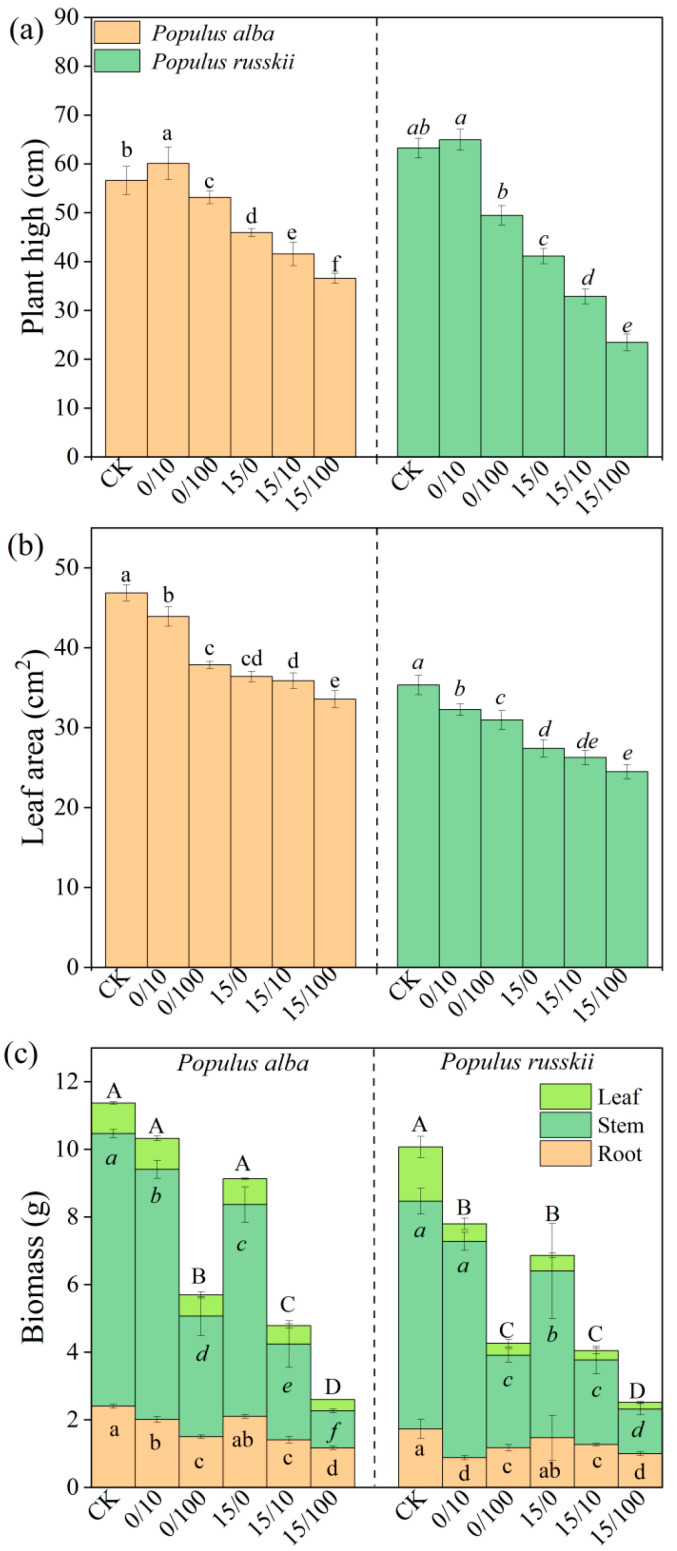
Growth responses of *P. alba* and *P. russkii* to different strontium and diesel oil treatments including plant height (**a**), leaf area (**b**) and biomass (**c**). Treatments are shown as oil/Sr combinations. CK means zero oil and Sr added. Different small letters above the bars (means of five replicates ± SE) indicate significant differences from one other at *p* < 0.05. Capital letters in figure C indicate significant differences in total biomass among treatments.

**Figure 2 plants-12-02145-f002:**
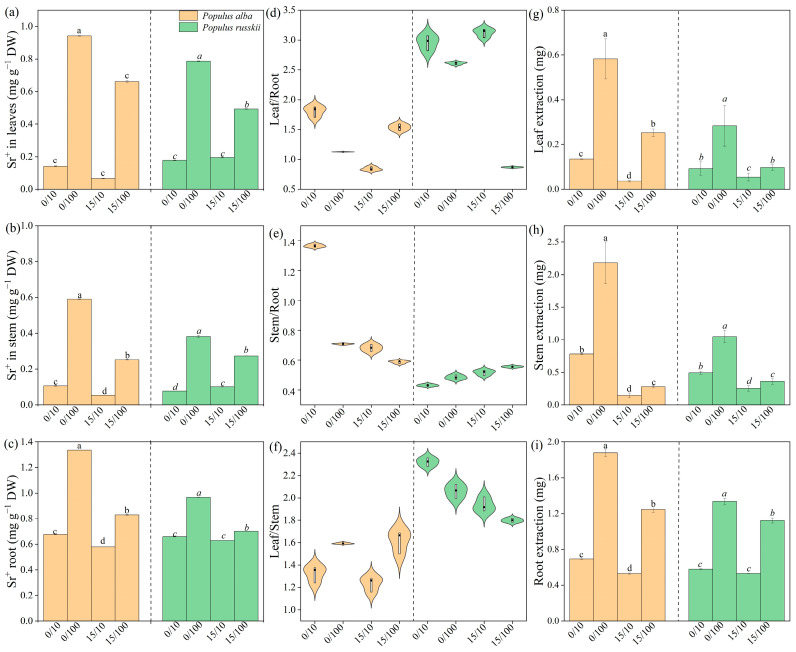
Strontium concentration (**a**–**c**), enrichment index (**d**–**f**), and extraction amount (**g**–**i**) of *Populus alba* and *P. russkii* under different strontium and diesel oil treatments. The strontium concentrations in 0/0 and 15/0 are below the detection line and are thus omitted. Treatments are shown as oil/Sr combinations. Different letters above the bars (means of five replicates ± SE) indicate significant differences from each other at *p* < 0.05.

**Figure 3 plants-12-02145-f003:**
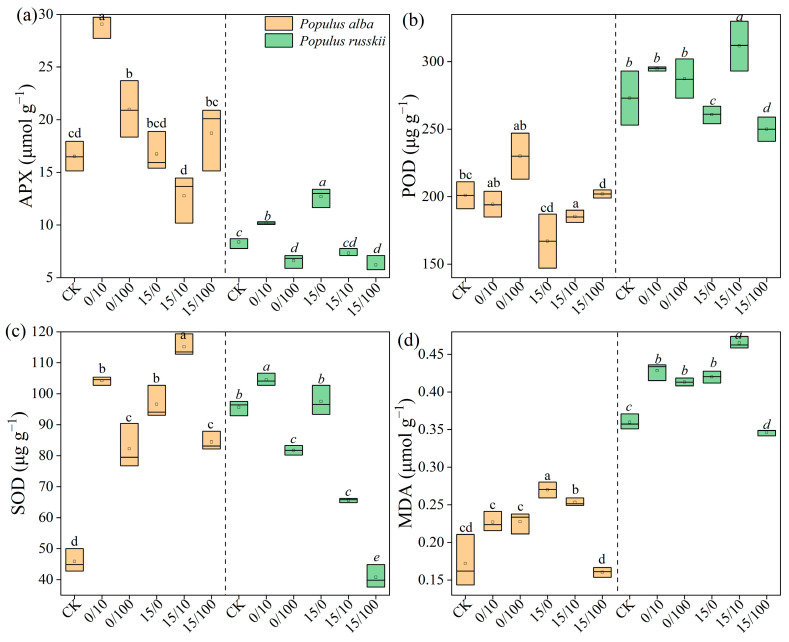
Antioxidant enzyme activities (APX, (**a**); POD, (**b**); SOD, (**c**)) and MDA contents (**d**) of *Populus alba* and *P. russkii* under different strontium and diesel oil treatments. Treatments are shown as oil/Sr combinations. Different letters above the bars (means of five replicates ± SE) are significantly different from each other at *p* < 0.05.

**Figure 4 plants-12-02145-f004:**
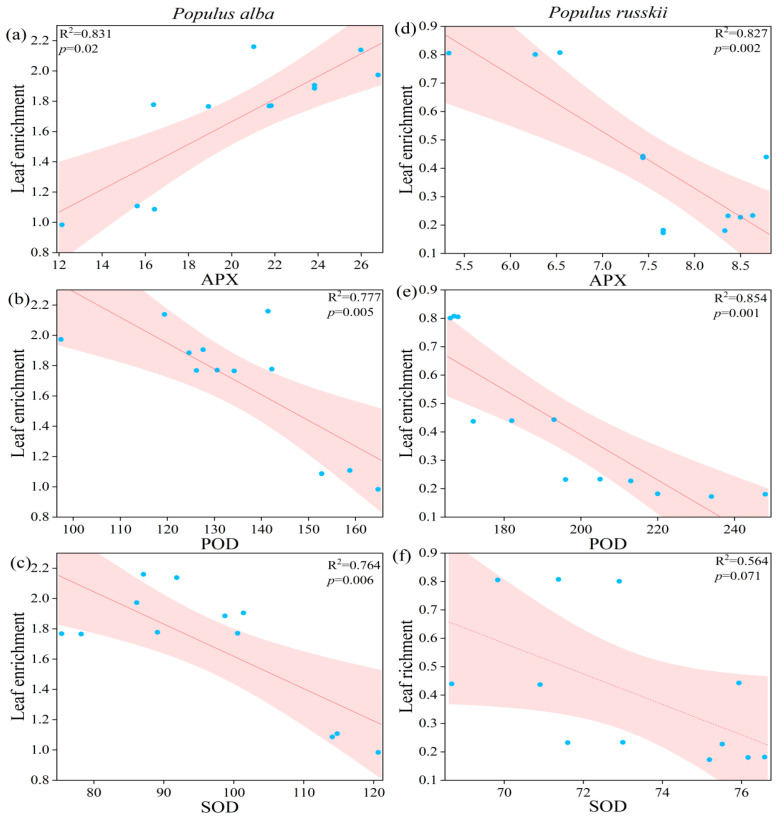
Correlation analysis between antioxidant enzymes (APX, (**a**,**d**); POD, (**b**,**e**); SOD, (**c**,**f**)) and leaf enrichment of *Populus alba* (**a**–**c**) and *P. russkii* (**d**–**f**) under different strontium and diesel oil treatments.

**Figure 5 plants-12-02145-f005:**
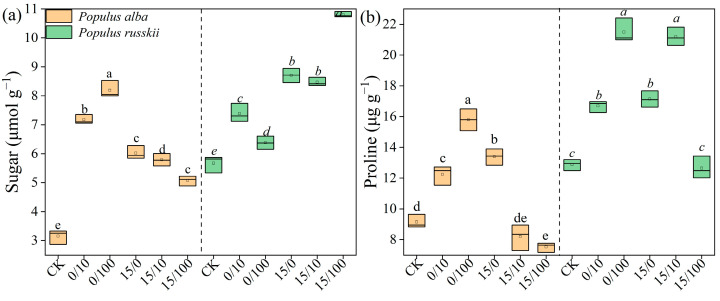
Contents of sugar (**a**) and proline (**b**) of *Populus alba* and *P. russkii* under different strontium and diesel oil treatments. Treatments are shown as oil/Sr combinations. The different letters above the bars (means of five replicates ± SE) are significantly different from each other at *p* < 0.05.

**Figure 6 plants-12-02145-f006:**
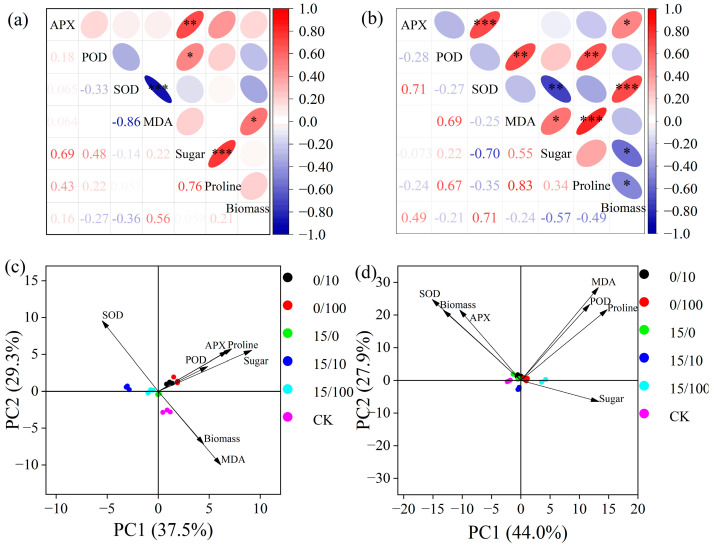
Contrasting physiological responses of *Populus alba* and *P. russkii* indicated in a heat map of the relationships among physiological indices (**a**,**b**); and principal component analysis (PCA) of the different treatments (**c**,**d**). * (*p* < 0.05), ** (*p* < 0.01), and *** (*p* < 0.001).

**Table 1 plants-12-02145-t001:** Summary of three-way ANOVAs evaluating the effects of species (*P. alba* vs. *P. russkii*), diesel oil (0 vs. 15 mg g^−1^), and Sr (0, 10, vs. 100 mg g^−1^) on growth indicators, enzyme activity, and soluble substances.

Factor	Enzyme Activity and Soluble Substances						Growth Indicators		
	APX	POD	SOD	MDA	Sugar	Proline	Leaf Area	Plant Height	Biomass
Species	<0.001	<0.001	<0.001	<0.001	<0.001	<0.001	<0.001	0.210	0.006
Oil	<0.001	<0.001	0.073	<0.001	<0.001	<0.001	<0.001	<0.001	<0.001
Sr	0.043	<0.001	<0.001	<0.001	<0.001	<0.001	<0.001	<0.001	<0.001
Species × oil	<0.001	<0.001	<0.001	<0.001	<0.001	<0.001	<0.001	<0.001	0.680
Species × Sr	<0.001	<0.001	<0.001	<0.001	<0.001	<0.001	<0.001	0.456	0.541
Oil × Sr	<0.001	0.003	<0.001	<0.001	<0.001	<0.001	<0.001	<0.001	0.123
Species × oil × Sr	<0.001	0.138	0.511	<0.001	<0.001	<0.001	<0.001	0.005	0.588

## Data Availability

The data presented in this study are available on request from the corresponding author.

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
