# Peer review of "Contrasting Responses and Phytoremediation Potential of Two Poplar Species to Combined Strontium and Diesel Oil Stress"

_plants, 2023, doi:10.3390/plants12112145_

Round 1
Reviewer 1 Report
The manuscript written by Liang et al. explored the phytoremediation potential of two poplar species on scrontium and diesel stress.
Overall, the research concept is scientific. Nevertheless, some corrections are required.
Specific comments:
1). The English language of the MS needs minor checks as some sentences lack appropriate punctuations. Please check throughout the MS.
2) What is the control in Figures 1? Also, only the Vertical and Horizontal lines should be shown in all the Figures.
3) The results in Tables 1 lack appropriate units
4) Improve the ‘Discusion’ section with the recent information/citations relating to the Sr and disease stress.
5) The links to the data information should be provided.
Minor comments.
L19 and 32: How many heavy metals?
L33: diesel oil? or diesel?
L46: areas2? Please check
L171-173: delete
L185, 188, 212, etc.: Avoid the use of ‘We’, rephrase. Check throughout the MS.
L266-267: Where exactly were the samples collected? Please indicate with appropriate information.
Minor checks are required.
Author Response
The manuscript written by Liang et al. explored the phytoremediation potential of two poplar species on strontium and diesel stress. Overall, the research concept is scientific.
Thank you for your positive comments.
Nevertheless, some corrections are required.
Specific comments:
1). The English language of the MS needs minor checks as some sentences lack appropriate punctuations. Please check throughout the MS.
The manuscript has been carefully edited (including grammar, spelling, punctuation, terminology use, sentence construction, readability, etc.) by MDPI English editing services (English editing ID: English-64927).
2) What is the control in Figures 1? Also, only the Vertical and Horizontal lines should be shown in all the Figures.
The control here was zero Sr and diesel added, i.e. 0/0. Yes, only the vertical and horizontal lines were shown in all the figures.
3) The results in Tables 1 lack appropriate units
Table shows the summary of three-way ANOVAs for the effects of species and treatment. So, there is no unit for the comparisons.
4) Improve the ‘Discussion’ section with the recent information/citations relating to the Sr and disease stress.
Yes. We have cited several recent references to improve the “Discussion” section.
Cheng, X.N.; Chen, C.; Hu, Y.M.; Guo, X.L.; Wang, J.L. Photosynthesis and growth of Amaranthus tricolor under strontium stress. Chemosphere, 2022, 308, 136234. doi: 10.1016/j.chemosphere.2022.136234
Chen, X.; Zhong, N.Y.; Luo, Y.Y.; Ni, Y.F.; Liu, Z.Y.; Wu, G.; Zheng, T.; Dang, Y.X.; Chen, H.L.; Li, W. Effects of strontium on the morphological and photosynthetic physiological characteristics of Vicia faba seedlings. Inter. J. Phytorem. 2022, 24: 811-821. doi: 10.1080/15226514.2022.2110037
Ren, H.; Huang, R.H.; Li, Y.; Li, W.T.; Zheng, L.L.; Lei, Y.B.; Chen, K. Photosynthetic regulation in response to strontium stress in moss Racomitrium japonicum L. Environ. Sci. . Pollu. Res. 2023, 30, 20923-20933. doi: 10.1007/s11356-022-23684-4
Zhang, C.Y.; Zhang, Z.K.; Zhou, J.H.; Wang, Y.; Ai, Y.M.; Li, X.P.; Zhang, P.J.; Zhou, S.B. Responses of the root morphology and photosynthetic pigments of ryegrass to fertilizer application under combined petroleum-heavy metal stress. Environ. Sci. Pollu. Res. 2022, 29, 87874-87883. doi: 10.1007/s11356-022-21924-1
5) The links to the data information should be provided.
Yes. We addressed “The data presented in this study are available on request from the corresponding author”.
Minor comments.
6) L19 and 32: How many heavy metals?
The following concentration (0, 10, or 100 mg kg-1) of Sr was added for the different treatments.
7) L33: diesel oil? or diesel?
It was diesel oil.
8) L46: areas2? Please check
It was changed into “area”.
9) L171-173: delete
Yes. That sentence has been deleted.
10) L185, 188, 212, etc.: Avoid the use of ‘We’, rephrase. Check throughout the MS.
Yes. These several sentences have been revised into passive voice.
11) L266-267: Where exactly were the samples collected? Please indicate with appropriate information.
Yes. We have added more information for the sampling site “samples were collected from Chengdu Botanical Garden (104°10’ E, 30°40’ N, 546 m a.s.l.) of Sichuan Province, Southwest China.
Reviewer 2 Report
Dear Authors,
your paper dealing with phytoremediation potential of two poplar species under combined stress could be of interest for the scientific community and I think is suitable for publication but i would like you to take into consideration some observation.
line 36: EU is going to stop the selling of vehicles running with diesel in 2035 so I would reconsider the whole sentence just considering the pollution present at the moment or increasing in the developing countries.
All the figures: you used CK and then 0/10 0/100 and so forth but in line 280 you introduced your treatments as T1 T2 T3...without CK. I suggest using the same indication of the treatments in the M&M and in the graphs.
line 114: I did not see the bold in the table.
Figure 2. There is not strontium at all in CK? Why did you delete CK from the graph? I would like you to also show the data of the control.
Line 171-177. Radioactive nuclide contamination is completely out of the aim of your work. I do not understand the purpose of these sentences here. It is a very diverse problem and Europe USA and China have completely different politics and sensibility about radioactive discharges. I would suggest changing or delete this part which is not useful for your paper.
line 255-257: If your paper is dealing with the double contamination is correct but suggesting that we could spill the oil in the soil on purpose is completely wrong in my opinion. Otherwise you should consider and measure and show how much of the oil you put in the soil remain as polluting. I think it is important to understand the cross tolerance and the differences among plants, but we cannot suggest to contaminate the soil with a toxic substance just to increase the uptake of another contaminant. Because we increase the total contamination of the site, or we introduce a contaminant that was not present at the beginning. I would cut out this suggestion which is impossible to accept in my opinion.
Author Response
Dear Authors,
your paper dealing with phytoremediation potential of two poplar species under combined stress could be of interest for the scientific community and I think is suitable for publication but i would like you to take into consideration some observation.
Thank you very much for your confirmation and further comments.
1) line 36: EU is going to stop the selling of vehicles running with diesel in 2035 so I would reconsider the whole sentence just considering the pollution present at the moment or increasing in the developing countries.
Yes. In the revised manuscript we have replaced “worldwide” with “developing countries”.
2) All the figures: you used CK and then 0/10 0/100 and so forth but in line 280 you introduced your treatments as T1 T2 T3...without CK. I suggest using the same indication of the treatments in the M&M and in the graphs.
We have changed them into thde same treatments indication in the text and figures.
3) line 114: I did not see the bold in the table.
That sentence has been deleted.
4) Figure 2. There is not strontium at all in CK? Why did you delete CK from the graph? I would like you to also show the data of the control.
The strontium concentrations in CK 15/0 treatments were below the detection line. So we delete them from the figures.
5) Line 171-177. Radioactive nuclide contamination is completely out of the aim of your work. I do not understand the purpose of these sentences here. It is a very diverse problem and Europe USA and China have completely different politics and sensibility about radioactive discharges. I would suggest changing or delete this part which is not useful for your paper.
Yes. That part has been deleted in the revised manuscript.
6) line 255-257: If your paper is dealing with the double contamination is correct but suggesting that we could spill the oil in the soil on purpose is completely wrong in my opinion. Otherwise you should consider and measure and show how much of the oil you put in the soil remain as polluting. I think it is important to understand the cross tolerance and the differences among plants, but we cannot suggest to contaminate the soil with a toxic substance just to increase the uptake of another contaminant. Because we increase the total contamination of the site, or we introduce a contaminant that was not present at the beginning. I would cut out this suggestion which is impossible to accept in my opinion.
Thank you very much for your insightful comments. That sentence has been revised into “Our study reported the existence of cross-tolerance to heavy metal and diesel oil stress, presenting a potential phytoremediation solution for combined toxicities.” We also revised the research aims accordingly.
